# MULTIPLE-FREQUENCIES POPULATION-BASED TRAINING

## ABSTRACT

Reinforcement Learning's high sensitivity to hyperparameters is a source of instability and inefficiency, creating significant challenges for practitioners. Hyperparameter Optimization (HPO) algorithms have been developed to address this issue, among them Population-Based Training (PBT) stands out for its ability to generate hyperparameters schedules instead of fixed configurations. PBT trains a population of agents, each with its own hyperparameters, frequently ranking them and replacing the worst performers with mutations of the best agents. These intermediate selection steps can cause PBT to focus on short-term improvements, leading it to get stuck in local optima and eventually fall behind vanilla Random Search over longer timescales. This paper studies how this greediness issue is connected to the choice of *evolution frequency*, the rate at which the selection is done. We propose Multiple-Frequencies Population-Based Training (MF-PBT), a novel HPO algorithm that addresses greediness by employing sub-populations, each evolving at distinct frequencies. MF-PBT introduces a migration process to transfer information between sub-populations, with an asymmetric design to balance short and long-term optimization. Extensive experiments on the Brax suite demonstrate that MF-PBT improves sample efficiency and long-term performance, even without tuning hyperparameters. Code will be released.

## 1 INTRODUCTION

The performance of neural networks depends on selecting a well-suited configuration of hyperparameters, a task that is time-consuming and often reduced to trial-and-error when done manually. This concern has driven the development of Hyperparameter Optimization (HPO, Bergstra et al. (2011); Feurer & Hutter (2019)), field focused on modeling and automating the hyperparameter selection process. The need for HPO algorithms is particularly strong in Reinforcement Learning (RL, Sutton & Barto (2018)), as RL algorithms are often highly sensitive to hyperparameter choices (Eimer et al., 2023; Zhang et al., 2021).

Given these challenges, Population-Based Training (PBT, Jaderberg et al. (2017)) has become a popular HPO method among RL practitioners (Badia et al., 2020; Liu et al., 2022). PBT trains a population of agents in parallel, using an evolutionary process to propagate successful hyperparameter configurations while exploring new ones. This frequent evolution enables PBT to generate dynamic hyperparameter schedules, unlike earlier methods like random search (Bergstra & Bengio, 2012) and classic sequential optimization (Li et al., 2018; Falkner et al., 2018), which typically produced fixed configurations. This dynamic adaptation of hyperparameters is particularly desirable in RL, where the learning problem is non-stationary (Parker-Holder et al., 2022).

However, to achieve this dynamic adaptation, PBT selects hyperparameter configurations based on intermediate performance. As a result, it often favors configurations that show early improvements but fail to deliver better long-term results. Dalibard & Jaderberg (2021) identified this greediness and proposed Faster Improvement Rate PBT (FIRE PBT), which uses learning curves to predict the long-term potential of hyperparameters based on their improvement rates. In this paper, we address PBT's inherent greediness by introducing a novel focus on evolution frequencies.

Evolution frequency, which controls the number of training steps between evolutionary updates, has not been explicitly addressed in prior research on PBT. Yet, our study shows that it lies at the core of a key trade-off in PBT's behavior. Evolving too frequently can lead to greedy collapse in two ways: (1)

aggressive hyperparameter tuning traps PBT in local optima, and (2) population diversity decreases as similar agents are reproduced repeatedly. Conversely, reducing the evolution frequency limits PBT's adaptability, resulting in less fine-grained schedules and, ultimately, deteriorating sample efficiency.

To address this trade-off, we propose Multiple-Frequencies Population-Based Training (MF-PBT), a novel HPO algorithm that employs multiple sub-populations, each evolving at distinct frequencies. By incorporating an asymmetric migration process, MF-PBT allows these sub-populations to share information while preventing greediness. This design aims to balance short and long-term optimization, leading to mixed-frequency schedules that enhance anytime performance.

We validate MF-PBT through a series of reinforcement learning experiments using the Brax framework (Freeman et al., 2021). Our results demonstrate that MF-PBT effectively mitigates the two forms of greedy collapse, achieving significantly higher long-term rewards and improved anytime performance compared to PBT baselines. Additionally, we conduct an empirical study on the potential of population-based methods for *variance-exploitation*, showing that even without hyperparameter tuning, populations can greatly improve performance by exploiting the inherent stochasticity of RL training. To ensure reproducibility, we make our code publicly available.

To summarize, our contributions are as follows:

1. We investigate the impact of evolution frequency on PBT and its connection to greediness.

2. We introduce MF-PBT, a novel HPO algorithm that uses multiple evolution frequencies and an asymmetric migration process across sub-populations to overcome PBT's greediness

3. We evaluate MF-PBT using the Brax suite, isolating the impact of PBT's greediness and demonstrating how MF-PBT mitigates this issue to achieve better final performance and sample efficiency across various environments.

4. We empirically show how population-based methods can leverage stochasticity in reinforcement learning training to significantly improve performance, even without explicit hyperparameter tuning.

## 2 PRELIMINARIES

Hyperparameter Optimization (HPO, Bergstra et al. (2011); Feurer & Hutter (2019)) encompasses various approaches aimed at efficiently tuning hyperparameters to enhance performance and robustness of learning algorithms. Random Search (Bergstra & Bengio, 2012) is the baseline approach to HPO; the field then progressed towards more sophisticated techniques, notably meta-gradient methods (Finn et al., 2017; Xu et al., 2018), sequential optimization (Li et al., 2018; Falkner et al., 2018; Awad et al., 2021), and population-based approaches.

In this section, we focus on Population-Based Training (PBT, Jaderberg et al. (2017)), beginning with its mechanisms and relevance to RL applications. Next, we briefly review notable extensions to PBT and provide insights into the greediness issue that we aim to tackle, highlighting its connection to evolution frequency.

### 2.1 POPULATION-BASED TRAINING

Population-Based Training (PBT) is an HPO technique that combines evolutionary strategies with gradient-based optimization. In PBT, a population of $N$ agents, $\mathcal{P} = \{a_i\}_{i=1}^N$, is trained iteratively in parallel, with each agent maintaining its own set of hyperparameters, $h_i$, and neural network parameters, $\theta_i$.

After every $t_{\text{ready}}$ training steps, an *evolution step* occurs where all agents are evaluated and assigned a fitness score. The parameter $t_{\text{ready}}$ controls the *evolution frequency*, with smaller values resulting in more frequent evolution. The agents are then ranked and divided into three brackets: *winners*, *survivors*, and *losers*. The evolution step consists of two phases: an exploitation phase, where the losers are replaced with clones of the winners, followed by an exploration phase, where the cloned agents' hyperparameters are slightly perturbed to encourage exploration around the best solutions. In our experiments, we use the *truncation* method introduced in PBT: the top $25\%$ are winners, the bottom $25\%$ are losers, and the remaining agents are survivors.

While PBT can be applied to any deep learning task, is is particularly effective in RL, due to the non-stationarity of the training process (Parker-Holder et al., 2022; Zhang et al., 2021). Unlike supervised learning, where the data distribution remains fixed, RL experiences significant shifts in the data distribution as training progresses, and the hyperparameters should take it into account. PBT's frequent evolution steps allow hyperparameters to adapt to the current learning state, naturally generating schedules that accommodate this non-stationarity.

Another strength of PBT is its ability to harness RL's intrinsic variance. The stochastic nature of both the environment and learning algorithms leads to significant performance fluctuations across different random seeds (Henderson et al., 2018; Agarwal et al., 2021). By maintaining a population and periodically reproducing the top performers, PBT propagates favorable outcomes, ensuring that unfortunate agents are replaced by luckier ones. This ability of PBT to propagate exploration luck is noted in Jaderberg et al. (2017), but our experiments in section 4.3 further demonstrate that population-based approaches can significantly improve performance, even without hyperparameter tuning.

Numerous extensions to PBT have been proposed, focusing on improving exploration and efficiency. Methods like PB2 (Parker-Holder et al., 2020; 2021) use bandit theory to explore hyperparameter spaces, offering performance guarantees, particularly in small population settings. SEARL (Franke et al., 2021) enhances sample efficiency in off-policy RL by employing a shared replay buffer across the population. BG-PBT (Wan et al., 2022) integrates policy distillation (Rusu et al., 2016) to jointly optimize neural architectures and hyperparameters.

However, these works do not address a key weakness of PBT: the inherent greediness of its intermediate selection mechanism. This issue was first identified in the original PBT work (Jaderberg et al., 2017), leading its authors to propose FIRE PBT (Dalibard & Jaderberg, 2021) to mitigate it through learning curve modeling. While FIRE PBT introduces an intricate mechanism (described in section 3.1), we aim to introduce a more practical approach of the greediness phenomenon.

All the aforementioned approaches rely on a fixed evolution frequency, and do not discuss the choice of the $t_{\text{ready}}$ parameter. To our knowledge, we are the first to investigate its impact on PBT, and its connection to greediness.

## 2.2 GREEDINESS AND EVOLUTION FREQUENCY

While PBT's dynamic adaptation of hyperparameters is a key strength, it also introduces a form of greediness in the optimization process. This greediness arises from selecting agents based on their short-term performance, often resulting in an overemphasis on immediate gains at the expense of long-term success. Evolution frequency lies at the core of this problem, as it controls the optimization horizon. Increasing $t_{\text{ready}}$ allows PBT to select agents based on longer-term performance, mitigating the short-sighted decisions issue. However, this comes at the cost of sacrificing PBT's main principle: its dynamic adaptation throughout the training run. We identify two collapse modes that can be caused by too frequent evolution: *diversity* collapse and *hyperparameter collapse*.

**Hyperparameter collapse.** Certain hyperparameters, such as the learning rate or exploration factors in RL, are inherently susceptible to greediness. Decaying these hyperparameters often yields immediate performance gains, making them more favorable during short-term selection. However, lower values restrict the exploration of the solution space, reducing the likelihood of finding better optima within $t_{\text{ready}}$ steps. This initiates a self-reinforcing cycle: agents with higher learning rates are outperformed and thus replaced by agents with lower learning-rates that fine-tune the found local optimum. After a few evolution steps, this hyperparameter collapse can combine with diversity loss, leading the overall optimization process to a convergence trap.

**Diversity collapse.** Diversity loss is a well-known weakness in evolutionary algorithms (EAs, Spears (1995)) that has not been directly addressed in PBT. When optimizing problems with multiple local optima, EAs often lose population diversity and converge to a single basin of attraction. Typically, this issue is corrected using niching techniques (Shir, 2012), which penalize reductions in diversity. In PBT, the repeated cloning of the highest-performing agents at each evolution step leads to a similar problem. Our variance-exploitation experiment in section 4.3 further highlights that this diversity collapse can cause PBT to fail, independently from hyperparameter optimization.

A solution to PBT's greediness should account for both of these collapses. One straightforward approach is to reduce the evolution frequency, giving agents more time to escape local optima and slowing the loss of diversity. However, this can't be a satisfactory solution, as it would directly harm PBT's sample efficiency by allowing poorly performing agents to persist longer. This ultimately pushed PBT closer to a Random Search, where evolution is entirely absent.

## 3 MULTIPLE-FREQUENCIES POPULATION-BASED TRAINING

To build upon our insights on evolution frequency, we propose MF-PBT, which employs multiple frequencies. By incorporating low-frequency agents that are less susceptible to hyperparameter and diversity collapse, alongside higher-frequency agents that enable quick adaptation and stronger anytime performance, MF-PBT mitigates the greediness of traditional PBT without sacrificing its core strengths.

### 3.1 SUB-POPULATIONS

A key challenge in PBT is the misalignment between short-term and long-term optimization. As the algorithm selects agents solely based on their performance over $t_{\text{ready}}$ training steps, and is blind to their long-term potential, it greedily favors hyperparameters that yield immediate gains, eliminating those that could lead to superior performance in the long term. Nevertheless, this short-term feedback is valuable to achieve strong anytime performance.

FIRE PBT (Dalibard & Jaderberg, 2021) introduced the concept of using sub-populations to address the trade-off between short-term and long-term optimization. In their approach, one sub-population is allowed to adopt a greedy strategy by directly optimizing the fitness signal, while the others aim to optimize a proxy for long-term performance: the *improvement rate*. To evaluate the long-term potential of hyperparameters, FIRE PBT uses an *evaluator* agent that simulates training with those hyperparameters. The core assumption is that faster improvement in the evaluator's performance indicates better long-term potential, which is a quite strong assumption on HPO.

In contrast, we argue that the best proxy for long-term performance is long-term performance itself. Rather than crafting an estimation, we let some agents train over longer timescales before evolution. In MF-PBT, each sub-population runs PBT at its own distinct evolution frequency. Dynamic sub-populations (i.e., higher frequency) focus on local optimization and short-term improvements, which can be greedy but offer gains in sample efficiency. Conversely, steady (low-frequency) sub-populations assess long-term performance, avoiding the pitfalls of greediness at the expense of sample efficiency.

Our main intuition comes from the phenomenon of greediness itself. When an algorithm shows strong early performance but eventually falls behind a simpler baseline, it is a clear sign of over-optimization and entrapment in a poor local solution. Based on this comparison principle, we expect dynamic agents to over-optimize local optima, and use the steady agents to regularly check if the dynamic agents have been greedy. Once greediness is identified, we correct it by restarting the optimization of dynamic agents around a better optimum found by steadier agents, a process managed through our asymmetric migration mechanism, details in next subsection.

### 3.2 ASYMMETRIC MIGRATION PROCESS

To effectively leverage the sub-populations, instead of running multiple PBT instances independently, an inter-population information transfer mechanism is needed. Alongside the winners, losers, and survivors brackets, MF-PBT introduces a migration bracket, allowing poorly performing agents within a sub-population to be replaced by better-performing agents from other sub-populations. The migration process operates asymmetrically based on the frequencies of the concerned sub-populations.

If a dynamic agent is outperformed by an agent from a steadier sub-population, this signals greediness. In response, we replace the dynamic agent with a clone of the steady one, to restore diversity in the dynamic sub-population and avoid convergence traps.

Conversely, if a steady agent is outperformed by a more dynamic agent, the dynamic agent's solution may result from a valuable high-frequency optimization pattern. However, since it might have been

achieved through over-optimization, we protect the steady sub-population from hyperparameter collapse by importing only the dynamic agent's weights, not its hyperparameters.

## 3.3 ALGORITHM

---
**Algorithm 1** Multiple-Frequencies Population Based Training (MF-PBT)

---
1: **procedure** TRAINING($\mathcal{P}$)
2:     **for** $\delta = 1, \ldots, T/t_{\text{ready}}$ **do**
3:         STEP($a, \forall a \in \mathcal{P}, t_{\text{ready}}$)                   $\triangleright$ Parallel Training for $t_{\text{ready}}$ steps
4:         $\mathcal{P} \leftarrow$ RANKING($\{a \in \mathcal{P}\}$)           $\triangleright$ Evaluate fitness and sort agents
5:         **for** $i = 1, \ldots, M$ **do**
6:             **if** $\delta \mod \delta_i = 0$ **then**                $\triangleright$ Population Update
7:                 $\mathcal{B}_i \leftarrow$ BRACKETS($\mathcal{P}_i$)
8:                 $\mathcal{P}_i \leftarrow$ EVOLUTION($\mathcal{P}_i, \mathcal{B}_i^1, \mathcal{B}_i^4$)
9:                 $\mathcal{P}_i \leftarrow$ MIGRATION($\mathcal{P}_i, \mathcal{P}_{-i}, \mathcal{B}_i^1, \mathcal{B}_i^3$)
10:             **end if**
11:         **end for**
12:     **end for**
13: **end procedure**

---

Similar to PBT, MF-PBT operates with a population of $N$ agents that train concurrently, evaluated every $t_{\text{ready}}$ steps and assigned a fitness score. The agents are divided into $M$ *sub-populations* $\mathcal{P}_1, \mathcal{P}_2, \ldots, \mathcal{P}_M$, each containing $n = N/M$ agents.

Each sub-population $\mathcal{P}i$ evolves at its own frequency, parameterized by the factor $\delta_i$, meaning it undergoes an evolution step every $\delta_i \times t_{\text{ready}}$ training steps. We set $\mathcal{P}_1$ to be the reference population, and the $\delta_i$ to be integers with $1 = \delta_1 < \delta_2 < \cdots < \delta_M$.

**Brackets.** When a sub-population $\mathcal{P}_i$ evolves, its agents are ranked and divided in four brackets: he top quarter, $\mathcal{B}_i^1$ (winners); the second quarter, $\mathcal{B}_i^2$ (survivors); the third quarter, $\mathcal{B}_i^3$ (open for migration); and the last quarter, $\mathcal{B}_i^4$ (losers). For simplicity, we assume $n$ is a multiple of 4.

**Evolution.** Regarding the winners, survivors and losers, MF-PBT behaves identically as PBT. The agents in $\mathcal{B}_i^4$ (losers) are replaced with perturbed clones of agents from $\mathcal{B}_i^1$ (winners). The survivors ($\mathcal{B}_i^2$) continue training unchanged.

---
**Algorithm 2** Asymmetric migration in MF-PBT

---
1: **function** MIGRATION($\mathcal{P}_i, \mathcal{P}_{-i}, \mathcal{B}_i^1, \mathcal{B}_i^3$)
2:     $k = 1$
3:     **for** $j = 1, \ldots, n/4$ **do**
4:         **if** FITNESS($\mathcal{B}_i^3(j)$) $\geq$ FITNESS($\mathcal{P}_{-i}(k)$) **then**
5:             **continue**       $\triangleright$ Agents in $\mathcal{B}_i^3$ better than contenders in $\mathcal{P}_{-i}$ are kept as is
6:         **end if**
7:         $i' \leftarrow$ INDEX($\mathcal{P}_{-i}(k)$)               $\triangleright$ Sub-population Index Retrieval
8:         **if** $\delta_{i'} < \delta_i$ **then**
9:             $\mathcal{B}_i^3(j)_\theta \leftarrow \mathcal{P}_{-i}(k)_\theta$            $\triangleright$ Weights Assignment
10:            $\mathcal{B}_i^3(j)_h \leftarrow \mathcal{B}_i^1(1)_h$          $\triangleright$ Hyperparameter Assignment
11:         **else if** $\delta_{i'} > \delta_i$ **then**
12:            $\mathcal{B}_i^3(j)_{\theta,h} \leftarrow \mathcal{P}_{-i}(k)_{\theta,h}$            $\triangleright$ Full Transfer
13:         **end if**
14:         $k \leftarrow k + 1$
15:     **end for**
16: **end function**

---

**Migration.** The agents in $\mathcal{B}_i^3$ are compared against agents in $\mathcal{P}_{-i} = \mathcal{P} \backslash \mathcal{P}_i$, to determine if they should be replaced by a copy of an external agent. First, both the agents in $\mathcal{B}_i^3$ and $\mathcal{P}_{-i}$ are sorted in descending order of fitness. Then, we sequentially perform pairwise comparisons of agents in $\mathcal{B}_i^3$ and

$\mathcal{P}_{-i}$. For each agent in $\mathcal{B}_i^3$, if it is outperformed by the current top external agent, we replace it using the asymmetric logic described in section 3.2. The procedure is detailed in Algorithm 2.

## 4 EXPERIMENTS

While MF-PBT can be applied to any HPO problem, we focus on reinforcement learning, where its impact is likely most significant. Following Wan et al. (2022), we use the parallelizable Brax framework (Freeman et al., 2021) to train a *Proximal Policy Optimization* (PPO) (Schulman et al., 2017) agent on multiple control tasks.

We use `jax`-based (Bradbury et al., 2018) implementations of MF-PBT and PPO, designed to parallelize agents on GPUs, thereby leveraging the capabilities of the Brax framework. This implementation achieves approximately $10^6$ steps per second on two Nvidia A100 40 GB GPUs, allowing us to train over extended timescales and clearly demonstrate PBT's limitations in the long term. For robust and fair evaluations, we conduct experiments on seven random seeds and report the interquartile means (IQM) (Agarwal et al., 2021) and interquartile ranges (IQR). To ensure reproducibility, we will make our code publicly available.[1]

We use a reference value of $t_{\text{ready}} = 10^6$ environment interactions, consistent with BG-PBT's experiments (Wan et al., 2022). This choice allow us to demonstrate how a conventional value can lead PBT to collapse over extended timescales. For the computation of the fitness score, we evaluate agents on 512 episodes and use the mean evaluation reward. Based on preliminary experiments, we selected $N = 32$ agents split into $M = 4$ sub-populations of $n = 8$ agents each, as moving from 16 to 32 agents significantly improved performance, while gains diminished beyond 32. In this setting, our longest experiments (3 billion steps in the *Humanoid* environment) require approximately 30 hours using two Nvidia A100 40 GB GPUs.

Our computational budget allowed us to train for approximately 1 billion steps per experiment, guiding our choice of $\delta_4 = 50$ for the steadiest sub-population. Indeed, higher values would get it closer to a random search, as the total number of evolution steps for this specific sub-population equals $^{1000}/\delta_4$. To facilitate smoother transfers between the fastest and slowest sub-populations, we selected two intermediary values: $\delta_2 = 10$ and $\delta_3 = 25$. This configuration of the $\delta$-values demonstrated slightly superior performance compared to a less spread geometric progression, as detailed in Appendix B.1. Given that the results already showed MF-PBT's ability to overcome PBT's greediness, we did not further tune the $\delta$-values.

We optimize the learning rate and the entropy cost of PPO's loss (Schulman et al., 2017), as these hyperparameters are particularly susceptible to causing hyperparameter collapse. For all experiments, we initially log-uniformly sample the learning rate between $10^{-5}$ and $10^{-3}$, and the entropy cost between $10^{-3}$ and $10^{-1}$. For the remaining hyperparameters, we use the tuned values proposed by Brax when available.[2] Notably, the same network architectures are used across all environments.

### 4.1 COMPARATIVE STUDY OF MF-PBT

We first compare MF-PBT to both PBT and Random Search (RS) (Bergstra & Bengio, 2012), using the same number of agents and the same value of $t_{\text{ready}}$. Since RS does not involve evolution, it can be viewed as a version of PBT with $\delta$ set to $+\infty$ (see Appendix A.1 for additional implementation details). This allows us to isolate the effect of evolution in PBT; if RS performs better than PBT, it is a clear sign of greediness.

For the perturbation of hyperparameters in both PBT and MF-PBT, we use the naive *perturb* strategy introduced in the original PBT (Jaderberg et al., 2017), which involves multiplying the hyperparameters by a factor $\lambda$ randomly sampled from $\{0.8, 1.25\}$. We do not include PB2 (Parker-Holder et al., 2020) and BG-PBT (Wan et al., 2022) as baselines, as these methods build on top of PBT to enhance exploration. Our focus is to identify and mitigate the inherent greediness in PBT's evolution mechanism. The improvements introduced in MF-PBT could potentially benefit PB2 and BG-PBT as well.

---

[1]The code will be published on GitHub after the double-blind review process. A minimal version of the project is included in the supplementary materials for reviewers.

[2]See Brax's GitHub. For *Hopper* and *Walker2D* we used the same values as in *Humanoid*.

FIRE PBT (Dalibard & Jaderberg, 2021) is also not included due to reproducibility challenges. It lacks a public implementation, and key aspects, such as the curve smoothing process, are not detailed in the paper. Additionally, their RL experiments use V-MPO (Song et al., 2019), an algorithm without a public implementation, and their experiment on ImageNet requires 200 TPU-v3 days, making direct comparison prohibitive.

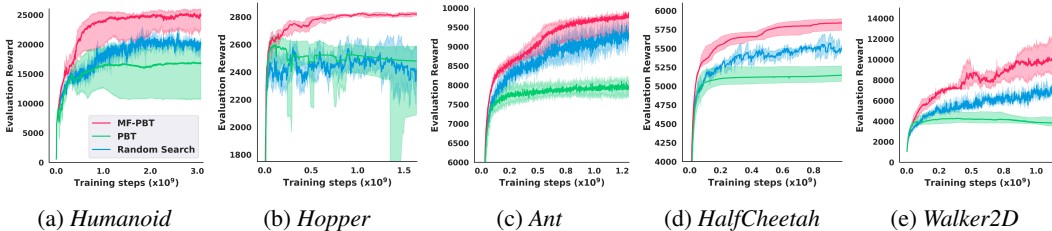

| (a) *Humanoid* | (b) *Hopper* | (c) *Ant* | (d) *HalfCheetah* | (e) *Walker2D* |

Figure 1: **Performance of MF-PBT, PBT, and RS on Brax environments.** IQM across seven seeds, with IQR shaded. The performance of each algorithm is determined by the highest fitness score (mean evaluation reward over 512 episodes) among the 32 agents, evaluated every $t_{\text{ready}}$ training steps.

This experiments highlight the limitations of regular PBT for long-term performance. In Table 1, we report the performance of each algorithm after 50 million training steps, the default horizon proposed by Brax for most environments. At this stage, PBT demonstrates relatively strong performance, achieving results that are superior or comparable to RS in most environments. However, after one billion steps, the same PBT falls significantly behind RS, demonstrating its greediness. Evolving every $t_{\text{ready}} = 10^6$ steps made it collapse in a poor optimum, while RS, which does not evolve, found better solutions.

In contrast, MF-PBT consistently outperforms both PBT and RS at both training horizons, showcasing its adaptability across varying timescales. The training trajectories reported in Figure 1 further illustrate that MF-PBT achieves stronger anytime performance, consistently outperforming RS and PBT throughout the training run. This indicates that MF-PBT has a better sample efficiency, achieving high rewards more rapidly.

Table 1: **IQM of the performance** achieved by the evaluated HPO algorithms at 50 million steps and 1 billion steps across seven random seeds. Methods within the IQR of the best-performing method are bolded. The *PPO* columns correspond to the training of a single agent with the default hyperparameters.

|  | Performance at 50M steps | | | | Performance at 1B steps | | | |
|---|---|---|---|---|---|---|---|---|
| Method | *PPO* | RS | PBT | MF-PBT | *PPO* | RS | PBT | MF-PBT |
| *Humanoid* | 7903 | **9021** | 8348 | **9266** | 14934 | 17713 | 16171 | **23793** |
| *Hopper* | 1782 | 2437 | **2542** | **2579** | 1822 | 2498 | 2519 | **2819** |
| *Ant* | 5482 | 6858 | 6820 | **7115** | 7102 | 9050 | 7900 | **9654** |
| *HalfCheetah* | 3786 | 4906 | 4914 | **5154** | 4262 | 5503 | 5143 | **5837** |
| *Walker2D* | 2881 | 3309 | **3822** | **3852** | 4261 | 7005 | 3870 | **9545** |

## 4.2 HYPERPARAMETERS SCHEDULES

To better illustrate MF-PBT's optimization process, we reconstruct the history of the best agent to visualize its hyperparameter schedule. In figure Figure 2a, we present three snapshots of MF-PBT taken during training on the *Humanoid* environment, at 750 million, 1.5 billion and 3 billion steps. For each snapshot, we trace back the history of the best-performing agent by recursively identifying the agents it cloned. Each colored segment in the schedule indicates the sub-population that produced the agent, showcasing how MF-PBT combines contributions from all sub-populations to produce its final solution.

A comparison of the three snapshots shows how MF-PBT is able to target for strong anytime performance. At every stage of the training, there are greedy agents diving into local optima, in order to maximize the immediate reward. Steady agents on their side, focus on long-term performance and protect the overall optimization process from collapse.

In the final schedule, we observe three phases. First, MF-PBT identifies an interesting high-frequency optimization pattern, where the learning rate increases briefly before decreasing, resembling the warm-up strategy proposed in Smith (2017). Next, the steady agents, slowly decrease their learning rate, avoiding collapse and aiming for better long-term rewards. Finally, dynamic agents take the lead, by fine-tuning the found local optimum through more aggressive learning rate decrease. This final schedules shows how MF-PBT effectively makes use of its multiple frequencies to produce the best long-term performance.

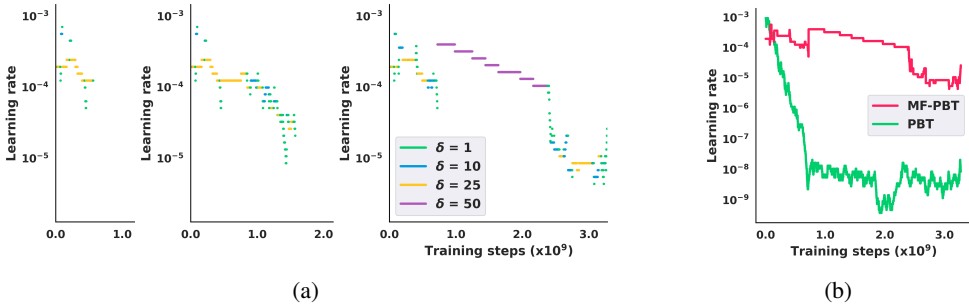

(a)                                                                                     (b)

Figure 2: **Example of learning rate schedules** for MF-PBT and PBT on the *Humanoid* environment. (a) MF-PBT snapshots at 750 million, 1.5 billion, and 3 billion training steps. Colors represent the sub-populations contribution to the schedule, showing how MF-PBT integrates input from various frequencies. (b) Comparison of the two final schedules, illustrating a case of hyperparameter collapse in PBT.

Figure 2b compares the final schedule produced by MF-PBT, to a schedule from a PBT experiment that encountered a strong hyperparameter collapse, ceasing to improve its reward after only 340 million steps. This collapse results from the presence of strong, peaked local optima in the *Humanoid* environment, such as running on one leg. Escaping such optima requires extensive exploration, as deviating from them is highly punitive, leading short-sighted PBT to enter a collapse cycle without finding better solutions. This difficulty with the *Humanoid* environment has also been noted in BG-PBT (Wan et al., 2022).

## 4.3 MF-PBT AS A VARIANCE-EXPLOITER

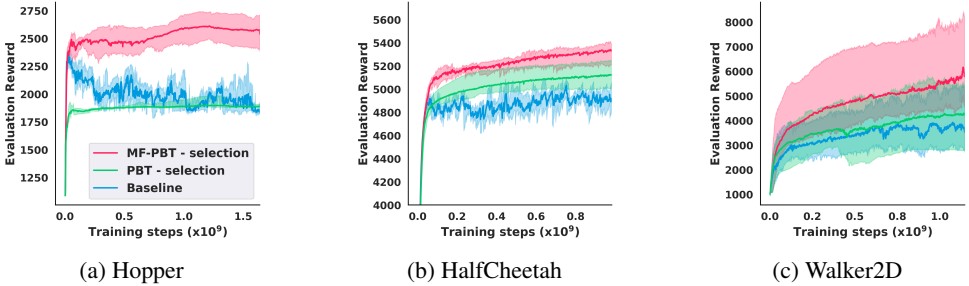

(a) Hopper                           (b) HalfCheetah                          (c) Walker2D

Figure 3: **Comparative performance of MF-PBT, PBT and a non-evolutive baseline for variance-exploitation.** IQM across seven seeds, with IQR shaded.

Building on our discussion on variance-exploitation in section 2.1, we designed experiments to evaluate MF-PBT's ability to leverage stochasticity in training outcomes to improve performance, even without hyperparameter tuning. In these experiments, all agents are fixed to use the default hyperparameters for the entire duration of training, with only weight cloning performed during the

evolution steps. To provide a baseline for comparison, we included a non-evolutive approach: running 32 agents independently without any weight replication or hyperparameter tuning, evaluating their fitness every $t_{\text{ready}}$ steps.

The resulting trajectories in Figure 3 reveal three key insights. (1) variance-exploitation can enhance the performance of a fixed hyperparameter configuration, as demonstrated in the *HalfCheetah* environment; (2) PBT, even when no hyperparameter collapse is possible, can still fall behind its non-evolutive counterpart, evidencing diversity collapse- the inherent greediness of the cloning mechanism; (3) MF-PBT significantly improves performance without modifying hyperparameters, illustrating the power of a more sophisticated cloning mechanism.

Interestingly, while PBT outperformed the non-evolutive baseline in the variance-exploitation regime for *HalfCheetah* and *Walker2D*, its performance dropped when hyperparameter tuning was introduced, indicating hyperparameter collapse. In contrast, MF-PBT performed in both regimes, highlighting its ability to overcome both diversity and hyperparameter collapse.

## 5   ABLATIVE STUDIES

### 5.1   EVOLUTION FREQUENCY

Our intuition is that evolving less frequently (increasing $\delta$) mitigates greediness and ensures better long-term performance, but using multiple frequencies is necessary to achieve stronger anytime performance. To test this, we conducted an experiment comparing MF-PBT with four separate PBT runs, each using 32 agents and evolving at one of the frequencies used within MF-PBT: $\delta \in \{1, 10, 25, 50\}$.

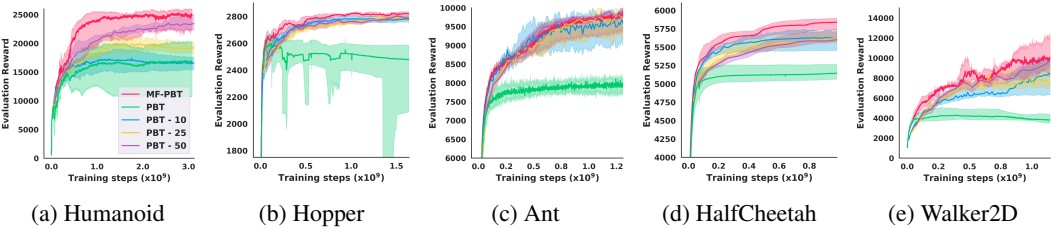

(a) Humanoid        (b) Hopper        (c) Ant        (d) HalfCheetah        (e) Walker2D

Figure 4: **Impact of the evolution frequency in PBT**. IQM across seven seeds, with IQR shaded.

The resulting trajectories plotted in Figure 4 confirm our first intuition about the critical role of evolution frequency, demonstrating its significant impact on PBT's performance. The curves also reveal that the best frequencies vary by task; for example, on *Humanoid*, $\delta = 50$ is the most effective, whereas on *HalfCheetah*, $\delta = 25$ yields better results. Additionally, most of the slower PBT configurations outperform RS, indicating that $\delta = +\infty$ is sub-optimal. This underscores the brittleness of population-based approaches to the choice of $t_{\text{ready}}$.

In contrast, MF-PBT achieves either superior or comparable final performance relative to each single-frequency PBT experiment, while also offering significant sample efficiency gains in most environments. This indicates that employing multiple frequencies within MF-PBT is superior to relying on a single frequency. Moreover, MF-PBT's ability to outperform each of its sub-components simplifies the selection of $\delta$-values, as MF-PBT will always perform at least as well as its best-performing sub-population.

### 5.2   SYMMETRIC MIGRATION

We now assess the importance of the asymmetry in the migration process, which adds a protection against hyperparameter collapse by preventing greedy agents from corrupting steadier sub-populations. To test this, we compare MF-PBT with an alternative version where hyperparameters are always transferred along with weights, regardless of the $\delta$-values.

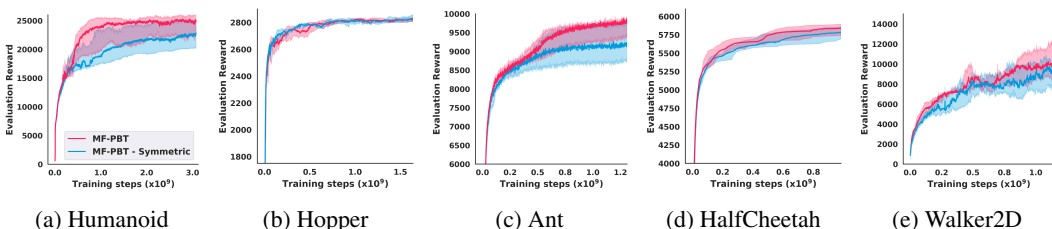

| (a) Humanoid | (b) Hopper | (c) Ant | (d) HalfCheetah | (e) Walker2D |

Figure 5: **Ablation on the asymmetric migration**. IQM across seven seeds, with IQR shaded.

The training trajectories in Figure 5 show that while the asymmetry has little impact on *Hopper*, it yields improvements in most environments, particularly in the challenging *Humanoid* task. This indicates that the asymmetric design indeed enhances long-term performance.

## 6 CONCLUSION

We introduced MF-PBT, an extension of Population-Based Training, designed to address the inherent greediness in traditional PBT. Our experiments on various reinforcement learning tasks identified two key failure modes of PBT: diversity and hyperparameter collapse, both linked to the evolution frequency. Building on these insights, we proposed MF-PBT, which incorporates multiple sub-populations evolving at different frequencies and an asymmetric migration process to balance short and long-term optimization. The results demonstrated that MF-PBT effectively overcomes both collapses associated with PBT while maintaining strong anytime performance.

Through ablation studies, we highlighted the critical role of evolution frequency in PBT and showed that using multiple frequencies increases robustness to this parameter. We believe this insight could be broadly valuable for all population-based approaches. Combining MF-PBT's mechanisms with other extensions to PBT, such as PB2 (Parker-Holder et al., 2020), PB2-Mix (Parker-Holder et al., 2021), or BG-PBT (Wan et al., 2022), could further enhance performance. Further work could also include a study of MF-PBT's performance on other tasks such as supervised learning.

Our experiments about variance-exploitation highlight that a non-negligible share of the performance gains in population-based methods arises from leveraging exploration luck rather than tuning hyperparameters effectively. This underscores the need for a more comprehensive study on the origins of improvements brought by population-based methods.

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

## A  Additional Implementation Details

### A.1  Random Search Baseline

In our experiments, Random Search (RS) serves as a simple baseline for hyperparameter optimization.
RS involves randomly sampling hyperparameter values at the start of training and keeping these
values fixed throughout the entire training process. Unlike PBT, RS does not involve any evolution
or adjustment of hyperparameters based on intermediate performance. Instead, the goal of RS
is to evaluate different fixed hyperparameter configurations by following their reward curves and
identifying which sampled configuration performs best.

For this comparison, hyperparameters in RS were sampled uniformly from the same search space as
PBT and MF-PBT. By comparing RS to PBT, we isolate the impact of PBT's evolutionary process;
if RS outperforms PBT, it indicates that evolving too frequently can lead to suboptimal long-term
performance, which we refer to as the greediness issue.

### A.2  PBT's parameters

In subsection 2.2 we identified that the main source of the greediness issue is that agents do not
survive long enough to escape poor local optima and maintain diversity. Alongside the evolution
frequency, another parameter of PBT impact the lifespan of agents in the population: the selection
rate in the exploit phase.

Indeed, in PBT, at each evolution step, $25\%$ of the population is discarded and replaced by copied of
the top-agents. One could play on this parameter to mitigate greediness, and create a method similar
to MF-PBT, where each sub-population would have its own selection rate. However, as we identify
the issue to be about the lifespan of agents, and optimizing for various horizons, we found more
natural to frame it explicitly in terms of evolution frequency.

We decided to use standard values for the *exploit* and *explore* process of PBT, and keep the same
values for MF-PBT in order to isolate the impact of evolution frequency.

## B  Choice of parameters

### B.1  Frequencies

Figure 6 compares our chosen configuration ($t_{\mathrm{ready}} = 10^6, \delta_1 = 1, \delta_2 = 10, \delta_3 = 25, \delta_4 = 50$) with
an alternative setup using a geometric progression ($t_{\mathrm{ready}} = 6 \times 10^6, \delta_1 = 1, \delta_2 = 2, \delta_3 = 4, \delta_4 = 8$).
The goal of this comparison is to assess how the spread of $\delta$-values impacts MF-PBT's performance.

While the geometric progression shows a slight advantage on *Hopper* and *Walker2D*, it performs
significantly worse on *Humanoid*. Therefore, we opted to continue using the more spread-out
configuration.

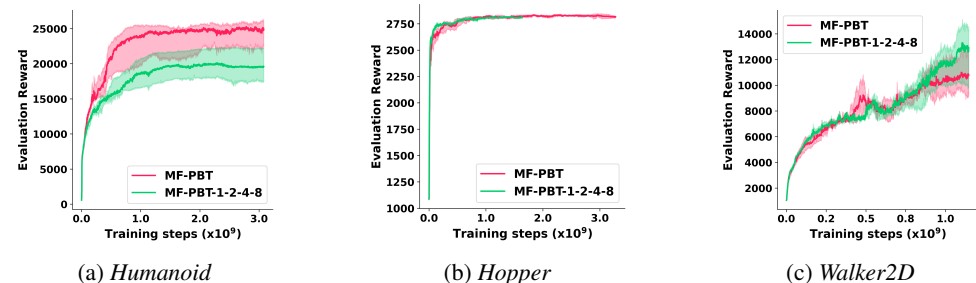

(a) *Humanoid*  (b) *Hopper*  (c) *Walker2D*

Figure 6: **Comparative performance of two configurations for MF-PBT.** IQM across seven seeds, with IQR shaded.

## B.2 POPULATION SIZE

To make a choice for $N$ after fixing the $\delta$-values, we conducted a preliminary experiment on *Humanoid*, the most computationally demanding environment. As shown in Figure 7, the gain from rising from $N = 16$ to $N = 32$ is quite large for both methods. While increasing from $N = 32$ to $N = 64$ was still beneficial for MF-PBT, but the with a much smaller gap.

Interestingly, PBT's performance decreases with 64 agents on the *Humanoid* task, likely due to the abundance of local optima. With a large population, PBT may quickly converge on a high-performing local optimum, which then limits further exploration.

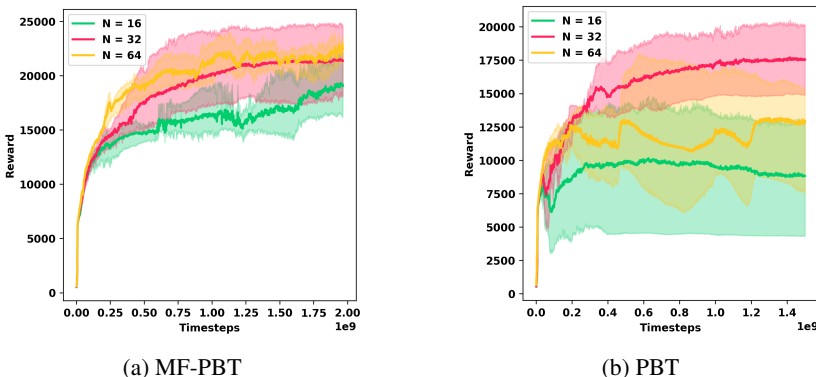

(a) MF-PBT  (b) PBT

Figure 7: **Impact of the population size.** IQM across five seeds, with IQR shaded. Experiments on the *Humanoid* environment.

## C ADDITIONNAL EXPERIMENTS

### C.1 PUSHER ENVIRONNMENT

We made an experiment in the *Pusher* environment from Brax, keeping the same parameters for MF-PBT, PBT and RS and report the training curves in Figure 8

### C.2 INCREASING POPULATION SIZE

One solution to improve PBT's performance can be to increase the population size. In Jaderberg et al. (2017), a value of $N = 80$ was used. To make sure we didn't unfairly treat PBT by picking $N = 32$, we made an additional experiment to compare the gains of using MF-PBT to the gains of simply increasing $N$ in standard PBT.

The curves in Figure 9 show that while on *Hopper* raising to 80 agents greatly improves PBT's performance, it is not sufficient in a more complex locomotion environment like *Walker2D*.

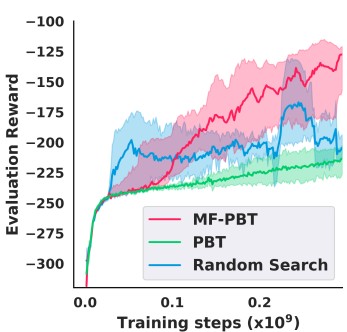

Figure 8: **Performance of MF-PBT, PBT, and RS on Pusher.** IQM across seven seeds, with IQR shaded.

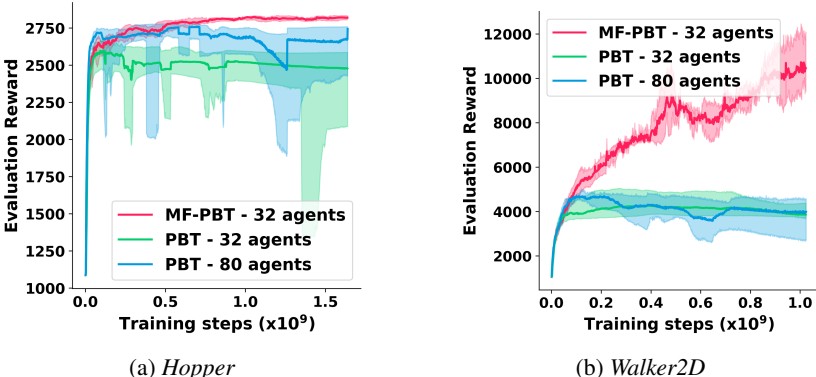

(a) *Hopper*                                    (b) *Walker2D*

Figure 9: **Increasing population size.** IQM across seven seeds, with IQR shaded.

## C.3 BACKTRACKING

Zhang et al. (2021) proposed to add a backtracking mechanism to PBT, to prevent it from catastrophic forgetting. The method, dubbed PBT-BT (PBT with backtracking), keeps track of the $N_e$ best agents encountered during the training: the *elites*. And every $\delta$ evolution steps, the elites are reincorporated into the population.

Since in the *Hopper* and *Humanoid* environments, we observed a substantial amount of runs where PBT's performance would dramatically drop, PBT-BT could be an interesting alternative baseline in those environments.

The backtracking can be seen as a migration across times, where elites from the past are reincorporated in the population, to enable it to resume training from a better checkpoint. However there is one fundamental difference, in PBT-BT the elites come from the past and didn't interact as much with the environnement; whereas in MF-PBT the steady agents that migrate are "current" agents, meaning they performed the exact same amount of training steps. In MF-PBT, the agents that migrate only differ on their HPO-objective, e.g. performance on 50M steps instead of performance on 1M steps. While backtracking enables recovering from collapses, there is no notion of increasing the lifespan of some hyperparameters to assess their long-term performance.

We implemented PBT-BT with $N = 32$, $N_e = 16$ and $\delta = 50$. The training curves in Figure 10 shows that it improves PBT on *Hopper* by correcting the catastrophic forgetting behavior. However on *Humanoid*, the elites tend to rapidly all belong to the same local optimum, and then PBT-BT is stuck without being able to explore for better solution.

In both cases, MF-PBT outperforms PBT-BT, highlighting that backtracking is not sufficient to overcome PBT's greediness.

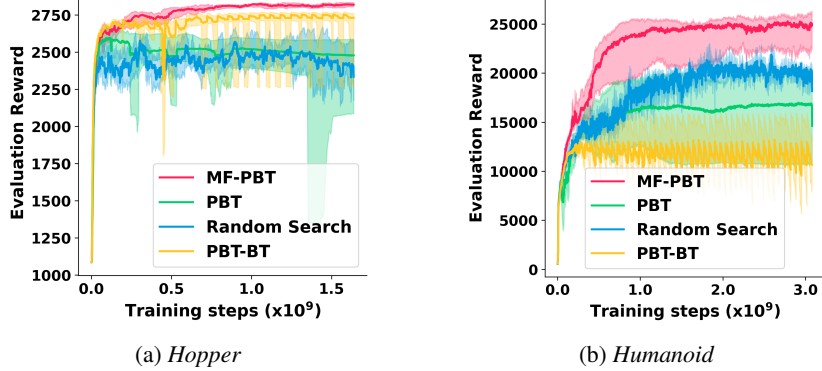

(a) *Hopper*

(b) *Humanoid*

Figure 10: **Comparative performance of PBT-BT.** IQM across seven seeds, with IQR shaded.

