# OpenReview forum: "Multiple-Frequencies Population-Based Training"
_ICLR.cc/2025/Conference — ICLR 2025 Conference Withdrawn Submission_

### Official Review · Reviewer_HjG6 · 2024-11-03

**Soundness:** 2
**Presentation:** 2
**Contribution:** 2
**Rating:** 6
**Confidence:** 5

**Summary:**

The work introduces an extension to population based training style hyperparameter optimization and evaluates the approach for reinforcement learning applications.

**Strengths:**

The work addresses an inherent issue of population based training style hyperparameter optimization. It clearly communicates the cause of these shortcomings and uses them to motivate the proposed changes to the PBT regime.
The work aims to provide an extensive empirical analysis based on the speedups gained by the Brax framework.

**Weaknesses:**

The work does not do a good job showing that their proposed sub-population scheme actually serves to improve the PBT training regime. In particular, the work identifies the frequency with which PBT calls the exploit-explore/mutation step as the root cause of PBTs inherent greedyness. However, I believe this ignores interaction effects of multiple hyperparameters in the PBT regime. First and foremost, the population size and the selection criteria play a crucial role. With a much larger population (such as 80 members, as suggested in the original PBT publication) the truncation method has a much larger pool of "survivor" members, which can keep constant hyperparameter settings. Similarly, if the fraction of "winners" and "losers" is reduced from 25% to, e.g., only 10 or even 5% then a similar effect is observed. This, essentially, leads to different frequencies with which the members are modified/replaced. Further, these two hyperparameters of PBT, joint with the "t_ready" flag have a huge impact on the overall training setting and are closely linked on how the overall training will progress. Out of these hyperparameters, only the "t_ready" has been studied in an ablation to show the impact on PBT, without adapting the population size of PBT. In this ablation it is clearly shown that PBT can reach similar good performances as the proposed modification even if the other hyperparameters are not further tuned. It is not unlikely that, with a bit of meta-tuning, PBT would achieve similar (or even better) performances than the proposed MF-PBT. As I could not find any mention on how sensitive MF-PBT is to its own hyperparameter settings (besides larger population sizes than 32 being hurtful), I am doubtful that MF-PBT is as much of an improvement as presented in the paper. However, due to the choice of a population size of 32, I am fairly certain that PBT underperformed. The recommendation of the original authors was to use at least a population of size 40 and reported best results for larger population sizes, such as 80.
I was surprised to see a huge number of validation episodes (512) to compute the fitness scores. This number is typically tried to be kept much lower since it adds a huge evaluation cost but many episodes are used to give an accurate fitness evaluation. I am curious if MF-PBT only performs well as it had access to so many evaluation episodes (something that is far from possible in more real world settings).

As the work decided to evaluate for excessively long training horizons which go far beyond the recommended training regime on the Brax environments, it can show large improvements compared to the baseline PBT. However, as reported in Table 1, the difference is not as huge on the standard training regime. It would have been better to use the available budget to compare against other methods such as PB2 or BG-PBT (with and without the multi-frequency adaptation) to show the actual impact of using the sbupopulations for with multiple frequencies.

I believe an important baseline is missing. PBT-BT (PBT with backtracking) as introduced by Zhang et al. 2021 (already cited in the work) uses an "elites" population to backtrack once PBT is stuck in a local optimum. To my understanding, this is done to better explore the hyperparameter search space at different steps in time. If I am not mistaken, this essentially also performs the "migration" across frequencies/time-steps as the population members can be replaced with earlier well performing members.
Further, since the work positions itself in the realm of HPO for RL it should also consider using other baselines common in this space (and not just static random search). Recently, multiple papers have shown that PBT (and PB2) are outperformed by classical HPO methods (see e.g. [Shala et al. 2022](https://openreview.net/forum?id=RyAl60VhTcG) [& 2024](https://openreview.net/forum?id=MlB61zPAeR), [Eimer et al. 2023](https://proceedings.mlr.press/v202/eimer23a.html) and [Becktepe et al. 2024](https://arxiv.org/abs/2409.18827?)) and in particular multi-fidelity approaches (commonly abbreviated MF due to which I would suggest a name change for MF-PBT). Counter to PBT style methods which potentially waste a lot of parallel compute to tune an agent, multi-fidelity approaches use partial trainings to explore the hyperparameter configuration spaces to quickly hone in on the best settings. Already the work by Zhang et al. 2021 that introduced PBT-BT showed that multi-fidelity approaches have complementary strenghts to PBT style tuning. As such, the work is incomplete without considering at least one multi-fidelity baseline or otherwise classical HPO baseline for the experiments.

Seven random seeds seems too low given that Brax has provide huge computational speedups.

The claim in the abstract that PBT can generate hyperparameter schedules in a single training run is false as PBT requires vast amounts of parallel resources and has already been called out in the SEARL paper which is already cited in the paper.

Overall, while the work proposes an interesting idea for PBT style HPO, I believe the experiments are missing crucial baselines and unfairly treat PBT. Thus my vote is reject.

**Questions:**

* Can you achieve a similar effect as your proposed MF-PBT by changing standard PBT hyperparameters?
* How does PBT-BTs backtracking compare to your migration strategy?
* How does your approach compare to other HPO for RL methods?

---

> ### Author Response · Authors · 2024-11-21
> **Regarding PBT's other hyperparameters**
>
> We thank you for your insightful and precise review, and really appreciate your sharp analysis of our work. We sincerely thank you in advance for taking the time to consider our rebuttal.
>
>
> > Can you achieve a similar effect as your proposed MF-PBT by changing standard PBT hyperparameters?
>
> Regarding your first point, you are right to say that both population size and selection criteria play also an important role in the greediness issue of PBT. The main matter being "do enough agents survive long enough with constant hyperparameters?".
>
> This being said, increasing population size isn't always the solution, as it becomes quickly expensive in terms of computing resources. We made a choice to stick with 32 agents as the performance increase from switching from 32 to 64 agents wasn't that big, and we indeed consider that 32 is already a large number and could be regarded as a limitation of both PBT and MF-PBT.
>
> To illustrate this point on population size, we added in the appendix of the paper, the curves from the experiments with 64 agents on Humanoid that guided our choice of using 32 agents.
>
> To also defend our work against the accusation of unfairly treating PBT, we performed additional experiments with 80 agents on Hopper and Walker2D. On Hopper, increasing the population size brought PBT's performance closer to that of MF-PBT (with 32 agents). However, on the more complex Walker2D task, there remains a significant performance gap between PBT 80 agents and MF-PBT 32 agents.
>
> Also on the general point that we didn't tune PBT enough, we made the choice to tune it just as much as MF-PBT. The same population sizes, selection rates, and mutations were used, and we sticked to common values introduced in the PBT paper. The only thing that differs between the two is the evolution frequencies, and our ablation in Figure. 4 is here to fairly indicate that, when the t_ready parameter is modified, PBT can perform way better.
>
>
> Finally, concerning the selection rate, we believe reducing the selection rate, and for example setting it as a probability to get replaced for "losers" agents, would have more or less the same effect as increasing t_ready. Therefore, it would be possible to make a version of MF-PBT with a fixed t_ready and one selection rate per sub-population. However, as we identify the issue to be about the lifespan of agents, and optimizing for various horizons, we decided to frame it explicitly in terms of evolution frequency.

---

> > ### Author Response · Authors · 2024-11-21
> > **Regarding PBT-BT**
> >
> > > How does PBT-BTs backtracking compare to your migration strategy?
> >
> > We must acknowledge that we missed PBT-BT, as it wasn't considered as a baseline in other PBT papers, probably due to the fact it didn't outperform PBT in Zhang et al.'s study. However regarding your comment, it appears to be a highly relevant baseline, though it does not have the same purpose as MF-PBT. To our understanding, the aim of backtracking is to recover from "catastrophic forgetting", by allowing agents to restart
> > from the best configurations (weights and hyperparameters) found in training. And it makes PBT even more greedy to avoid collapses, while our objective is to avoid greediness.
> >
> > We observed in the Hopper and Humanoid environments that PBT's performance would often dramatically drop, due to all the population collapsing in unstable optima (e.g. single-leg running). In those cases PBT-BT could be a stronger baseline than PBT, by frequently reincorporating the elites agents.
> >
> > The backtracking can be seen as a migration across times, where elites from the past are reincorporated in the population, to enable it to resume training from a better checkpoint. However there is one fundamental difference, in PBT-BT the elites come from the past and didn't interact as much with the environnement; whereas in MF-PBT the steady agents that migrate are "current" agents, meaning they performed the exact same amount of training steps. In MF-PBT, the agents that migrate just differ on their HPO-objective, e.g. performance on 50M steps instead of performance on 1M steps.
> >
> > While backtracking enables recovering from collapses, there is no notion of increasing the lifespan of some hyperparameters to assess their long-term performance. Hence, we don't believe PBT-BT to be able to escape poor local optima without playing on the t_ready parameter.
> >
> > To make a comparison between PBT, PBT-BT and our MF-PBT, we reimplemented PBT-BT with 32 agents, and keeping track of 16 elites, that are reincorporated every 50 evolutionary steps, the value 50 being chosen as the best performing frequency in our ablation studies. When an elite is reincorporated, it undergoes one mutation step to ensure the exact same hyperparameter that led to collapse won't be reused.
> >
> > We made the first experiments on Hopper and Humanoid, where we believe PBT-BT would be the most valuable, and report the associated curves in the appendix of the updated pdf. We see that while on Hopper PBT-BT indeed brings a clear improvement over PBT, it remains behind MF-PBT.
> >
> > On the highly complex Humanoid environment, with numerous local optima (running without using knee joints, single-leg...), adding backtracking doesn't improve on PBT, and even worsen it a bit, likely because all the elites quickly converge to the same local optimum. Zhang et al. already pointed PBT-BT as a "more greedy version of PBT" in the appendix of their paper.

---

> > > ### Author Response · Authors · 2024-11-21
> > > **On our framing of the HPO problem**
> > >
> > > > As the work decided to evaluate for excessively long training horizons
> > >
> > > Prior answering to your last question, we would like to rebound on this comment, as we believe it is a point of primary importance.
> > >
> > > To our understanding, the Bayesian Optimization (BOHB, DEHB, SMAC...) literature frames the HPO problem in the following way: given a fixed computational budget $B$, find the **best performing hyperparameter configuration** possible. And then, practitioners would reuse the found hyperparameters for their trainings. This is the setting of the Figure 1. of Zhang et al. 2021, where we see Hyperband to outperform PBT.
> > >
> > > Population-based methods (PBT, PB2, SEARL, MF-PBT...) frame the problem as find the **best performing agent possible**, and the budget rather consists of parallel computing abilities. This is the main reason why these papers don't present a study of the reusability of the schedules. This is the setting of the Figure 4. of Zhang et al. 2021, where PBT shows to perform better.
> > >
> > > Once we decided our objective is to find the best agent possible, we believe there is no point in comparing HPO methods after "only" 50M steps. We see that Random Search is able increase its performance for more than 1B steps in most environments, indicating there is still much room for improvement after the first 100M steps.
> > >
> > > Additionally, the main subject of our proposition is about greediness. To be able to show that we effectively mitigated this major issue of PBT, we must work on extended timescales, to be sure our method didn't get stuck in a poor local optimum.

---

> > > > ### Author Response · Authors · 2024-11-21
> > > > **Absence of BO baselines**
> > > >
> > > > > How does your approach compare to other HPO for RL methods?
> > > >
> > > > We believe our experimental setting to be quite unfair to standard BO-methods. As mentioned in section 4, we tune only two hyperparameters (learning rate and entropy coefficient), and all the remaining hyperparameters (gamma, batch sizes...) are taken for the already-tuned configurations proposed by Brax. This setting makes Random Search quite representative of the results achievable by methods yielding fixed hyperparameters configurations. The fact that MF-PBT performs better than the 7 seeds of 32-agents RS (Figure. 1), indicates it would probably also outperform BO methods (in this setting).
> > > >
> > > > Also we do not believe that PBT "wastes" computational resources, even if at the end of the training only one agent is outputted, the 31 others were necessary to its production. Our experiments on "variance-exploitation" in section 4.3 show how much using multiple agents can improve performance of RL trainings, even without tuning hyperparameters.
> > > >
> > > > To sum up, PBT-style and classical BO-based HPO are complementary, and don't follow the same objective. One way of using both could be for example to use SMAC or DEHB to first find a well-performing hyperparameter configuration, and then run MF-PBT on a smaller set of hyperparameters that highly benefit from dynamic tuning (mostly the learning rate) to further improve performance and achieve the best rewards.

---

> > > > > ### Author Response · Authors · 2024-11-21
> > > > > **Additional Points**
> > > > >
> > > > > Alongside the questions you asked, we would also like to answer to points you made in the Weaknesses section.
> > > > >
> > > > > > I was surprised to see a huge number of validation episodes (512) to compute the fitness scores
> > > > >
> > > > >
> > > > > We indeed acknowledge that it is quite a large number, but using Brax makes it affordable. Our motivation behind this choice is that we wanted to get rid of evaluation variance in our curves, so the IQR spread only characterizes the variance of the HPO methods. Moreover, in a setting where training 32 agents for 3B steps is affordable, we believe it is a reasonable assumption that selecting agents based on a accurate estimation of their performance is also affordable.
> > > > >
> > > > > Since we used the same number of evaluation episodes for all methods, it is quite unfair to imply we made this choice to favor MF-PBT. If it is a blocking point for you, we can make an additionnal experiment with less episodes in the fitness computations, but we are convinced it would impact PBT and MF-PBT in the same way.
> > > > >
> > > > >
> > > > >
> > > > >
> > > > >
> > > > > > Seven random seeds seems too low given that Brax has provide huge computational speedups.
> > > > >
> > > > > We made the choice to use the Brax's speedups to perform extended trainings, most RL papers report training curves that didn't reach convergence, which we believe isn't desirable as final performance is more important in real-world settings (you want the best possible model to be put in production).
> > > > >
> > > > > In this setting, obtaining seven random seeds for all experiments including ablations is already quite expensive in terms of both time and computing resources.
> > > > >
> > > > >
> > > > > > The claim in the abstract that PBT can generate hyperparameter schedules in a single training run is false
> > > > >
> > > > > It is true that PBT requires vast amounts of parallel ressources, but we do believe that in terms of wall-clock time it is still correct to say that PBT makes uses of a single run to produce a schedule. However, we understand that this claim is misleading as it supposes labs to be able to run 32 trainings in parallel, and that since PBT went out in 2017 HPO research made a lot of progress towards efficient algorithms such as DEHB.
> > > > >
> > > > > We propose to keep the sentence in the abstract because it is indeed one of the main arguments that made PBT a popular approach, but we're open to modify it if you insist such a misleading claim shouldn't appear in our abstract.
> > > > >
> > > > >
> > > > > To conclude, we once again greatly appreciate your review, and your comments really brought us up with fruitful reflections on our work. We hope to have addressed your most critical points, and are looking forward pursuing the discussion with you.

---

> ### Comment · Reviewer_HjG6 · 2024-11-25
> **Response to rebuttal**
>
> Thank you very much for the thorough rebuttal. In the mean time I have read all other reviews and rebuttals. **I increase my score to 6.** While not all of my concerns have been resolved, I better understand the framing the authors see for their work and am willing to concede some of my criticism in favor of the authors (thus increasing the score to 6 instead of 5).
> My biggest gripe is still with regards to the experimental setup and I still believe that classical tuning methods are needed as baselines. It is still not well understood how much dynamic tuning is actually needed and tuned static configurations should therefor always be a considered baseline in PBT style settings. As stated before however, as the work does not claim to be the best tuning approach for AutoRL/AutoML, I am willing to accept that the authors experiments compare a dynamic approach to other dynamic approaches.
>
> Two final remarks:
> * I am still very much in favor of removing the claim of "single training run" in the abstract.
> * Adding the discussion from the rebuttal "Regarding PBT's other hyperparameters" and "Regarding PBT-BT" to the appendix could help other readers better understand the method since I also found it very instructive

---

> > ### Author Response · Authors · 2024-11-26
> > **PDF updated + additional thoughts on static vs dynamic tuning**
> >
> > We are deeply thankful that you considered our rebuttal and reevaluated your appreciation of our work.
> >
> > We took into account your two final remarks and updated the pdf accordingly. Notably, we have replaced "in a single training run" with "instead of fixed configurations".
> >
> > > It is still not well understood how much dynamic tuning is actually needed
> >
> > This is actually a very important point. Our experiments about variance-exploitation (section 4.3) show that even with fixed hyperparameters, population-based methods can greatly improve training performance. This result is noteworthy because prior literature often attributes the success of PBT solely to its dynamic adaptation capabilities. However, our findings suggest that the contribution of "variance-exploitation" is an important part of the gains of population-based methods.
> >
> > In this regard, comparing MF-PBT to a BO-baseline would not directly be a comparison of dynamic tuning vs static tuning; but more "dynamic tuning + variance-exploitation" vs. static tuning, and might once again be unfair to the BO baseline.
> >
> > A possibity to isolate the impact of dynamic tuning in the PBT framework, would be to compare a variance-exploiter PBT running with the static configuration produced with a BO baseline, to either directly PBT or a variance-exploiter PBT that re-uses a schedule found by PBT.
> >
> >
> > Another hint of the importance of the variance-exploitation ability of PBT lies in Table 1. of the paper. In this table, we see that even the poorly-performing greedy PBT outperforms the static config that runs with a single-agent on almost every environment. This raises a further question about comparing with a BO baseline in our setting: it wouldn't be fair to compare the training curve of a single-agent using the configuration found by the BO-baseline to the performance of a 32-agents experiment.

---

### Official Review · Reviewer_o539 · 2024-11-04

**Soundness:** 3
**Presentation:** 3
**Contribution:** 3
**Rating:** 8
**Confidence:** 4

**Summary:**

The paper presents Multiple Frequency Population Based Training (MF-PBT), a novel evolutionary algorithm for hyperparameter optimization. The method allows different individuals in the population to evolve at different frequencies, rather than having all individuals update at the same rate. The authors demonstrate that performing parameter migration without hyperparameter migration helps avoid hyperparameter collapse, a common issue in standard PBT approaches.

**Strengths:**

1. Novel and well-motivated idea of having multiple evolutionary frequencies within a population, the issue of hyperparameter sensitvity in Reinforcement Learning is well known and the paper makes a good attempt at addressing it.
2. Strong empirical validation showing benefits over standard PBT and a smart use of random search to make sure greediness is avoided
3. Important insight about separating parameter and hyperparameter migration to prevent collapse are substantiated and that is a useful understanding for the community
4. Fits well into existing literature on evolutionary dynamics in population-based RL methods
5. Motivated and detailed experimental setup and ablation studies

**Weaknesses:**

## Major Weaknesses
1. The evaluation and definition of "anytime performance" would benefit from formalization increasing the notion's rigor.
2. Current results don't fully demonstrate if better early performance is due to high-frequency evolution or other factors such as good performance in the beginnings of training.
3. Limited exploration of different frequency spreads and not given enough weight in the main results (only two tested in supplementary section which are important to explore)
4. Not clear if the findings generalize beyond PPO
5. Not clear how sensitive is MF-PBT to the choice of frequency spread. A more comprehensive analysis of different spreads would be valuable.

## Minor Weaknesses
1. Results section organization could be improved by showing performance results before hyperparameter analysis
2. The motivation for anytime performance could be better developed
3. While random search and PBT are good baselines, comparison to other hyperparameter optimization methods would strengthen the evaluation

**Questions:**

1. How does the computational overhead of managing multiple frequencies compare to standard PBT when there is an asymmetry between the cost of evolutionary updates and gradient updates?
2. Could adaptive frequency selection (rather than fixed frequencies) provide additional benefits?
3. How does MF-PBT perform on tasks with different types of problems such as sparse rewards or high dimensional action spaces?

---

> ### Author Response · Authors · 2024-11-24
> **On the notion of Anytime Performance + Answers to the questions**
>
> We would like to start by thanking you for reviewing our work and for your positive and encouraging feedback.
>
> In this response, we first address your comments on "anytime performance," as we recognize this as an important point that we may not have clarified sufficiently in the paper. Following that, we address your other questions in detail.
>
> **About Anytime performance**
>
> We acknowledge that our usage of "anytime performance" could be more clearly formalized in the paper, as we use it to describe two distinct but closely related ideas:
>
> 1. The idea that the training curve of a method consistently stays above the one of another method. For example, in figure 4 of the paper, we see that the PBT curves intersect: for the first 100M steps PBT-10 is better, but then it is PBT-50 that becomes the stronger alternative.  However, MF-PBT's curve remains above all PBT variants throughout the training run. This indicates that regardless of training horizon, MF-PBT is the better choice.
> 2. What we also call anytime performance, especially in the section 3 where we describe MF-PBT, is the ambition to achieve the highest possible scores at every stage of training. In figure 4a., while PBT-50 and MF-PBT converge to similar final performance levels, MF-PBT reaches that level in roughly one-third of the time required by PBT-50. This suggests that MF-PBT offers substantial gains in sample efficiency.
>
> The two notions are quite close, but one is like a term to use when comparing two methods, and the second is more about the ambition of the method.
>
> Formalizing the first definition is relatively straightforward (curve_a always over curve_b, or more than x% of the time).
>
> However formalizing the secondr equires access to an oracle capable of identifying the true optimal performance at each training step, which is unavailable in Brax and most other environments. Thus, we emphasize that MF-PBT is designed with the ambition of having a great anytime performance, and our experimental results demonstrate that it achieves greater sample efficiency than competing baselines.
>
>
> Thanks again for pointing this out, we will make a pass on the paper to add clarity on this notion. And we're open to discussion if you have further suggestions on this matter.
>
>
> > How does the computational overhead of managing multiple frequencies compare to standard PBT when there is an asymmetry between the cost of evolutionary updates and gradient updates?
>
> The main difference between MF-PBT and standard PBT happens during the evolution step, after all the agents are ranked.
>
> These steps just consist of ranking the agents and reassigning them new hyperparameters, so it is negligeable in front of the gradient updates and environment interactions. To this regard, one can say there is no computationnal overhead in using MF-PBT compared to using standard PBT.
>
>
> > Could adaptive frequency selection (rather than fixed frequencies) provide additional benefits?
>
> Since our work demonstrates the critical impact of evolution frequency on PBT's performance, we agree that an adaptive frequency selection mechanism could indeed provide benefits.
>
>
> However, it raises multiple questions. For example, to know if a frequency is "good" or should be modified, we need to compare its performance with another frequency.
>
> But we don't really want to increase the frequency (i.e. decrease t_ready), as we know that high-frequency populations perform better on the short-term before getting stuck in poor optima.
>
> Conversely, high-frequency populations contribute to local optimization, helping explain MF-PBT’s performance advantage over, for example, PBT-50 (Fig. 4 of the paper).
>
> Having adaptive frequency selection would require a proxy capable of determining—based solely on the current performance of a PBT training—whether the frequency should be adjusted. Our stance is that such proxies don't exist, or would rely on strong assumptions (cf. FIRE-PBT), hence our choice to use multiple-frequencies.
>
>
>
>
>
>
> > How does MF-PBT perform on tasks with different types of problems such as sparse rewards or high dimensional action spaces?
>
> Building up on our observations on Brax and on our intuition, we believe MF-PBT shows the greatest improvements in complex environments with multiple local optima such as *Humanoid* and *Walker2D*.
>
> So, in problems with multiple local optima, which can happen with higher dimensional action spaces, MF-PBT should bring improvements over PBT by mitigating its greediness.
>
> However, we do not have an intuition about the correlation between sparse rewards and the presence of multiple local optima, so we are unable to draw a conclusion.
>
> Thank you once more for your review, we would welcome the opportunity to further discuss if some points still need to be clarified.

---

> > ### Comment · Reviewer_o539 · 2024-11-25
> > **Anytime Performance**
> >
> > Maybe the notion of anytime performance you're looking for is similar to the notion of regret in Reinforcement Learning. To make it practical, you can compare at each point in time the performance of a model compared to the best performance you found, rather than the most optimal that can possibly be found at each time step. This error from the optimal point can be used as a metric (regret) for comparisons across methods. As far as sample efficiency is concerned, average steps before reaching a threshold of performance is a good metric in my opinion. These are just suggestions, thank you for your in-depth response.

---

> > > ### Author Response · Authors · 2024-12-04
> > >
> > > Thank you for your valuable suggestions.
> > >
> > > The notion of regret you propose is indeed an interesting formalization of anytime performance. We did not have the time to conduct a thorough analysis of this metric, but we will consider adding it to the paper. We also need to reflect on whether it would be more meaningful to compare against the absolute maximum performance found (per run), or to the maximum of the IQM on seven seeds (per algorithm).
> > >
> > > Regarding sample efficiency, a challenge is that we did not know *a priori* what is a "good" performance on Brax, mostly because most paper work train on  ~100M steps. We did not want to set an arbitrary performance threshold, but we are convinced that, as Brax is becoming more used, the community will converge on standard thresholds, enabling this kind of comparisons.
> > >
> > > Thank you once again for engaging with our work and providing thoughtful feedback.

---

> ### Author Response · Authors · 2024-11-24
> **Generalization beyond PPO**
>
> > Not clear if the findings generalize beyond PPO
>
>
> >  How does MF-PBT perform on tasks with different types of problems such as sparse rewards or high dimensional action spaces?
>
> For both these points we would like to underscore that we didn't make assumptions on the nature of the problem, or on the RL algorithm. So we believe that MF-PBT would sill overcome PBT's greediness in other settings.
>
> Our only motivation behind working with PPO on Brax is the huge speed of Brax's official implementation of PPO, that enabled us to train on extended timescales, and perform all experiments and ablations on seven random seeds.

---

### Official Review · Reviewer_Xs8X · 2024-11-04

**Soundness:** 2
**Presentation:** 2
**Contribution:** 1
**Rating:** 3
**Confidence:** 4

**Summary:**

The paper introduces Multiple-Frequencies Population-Based Training (MF-PBT), a hyperparameter optimization algorithm addressing the greediness issue in PBT. By employing sub-populations that evolve at different frequencies and an asymmetric migration process, MF-PBT aims to balance short-term and long-term optimization. Experiments on the RL locomotion tasks demonstrate that MF-PBT improves sample efficiency and long-term performance, even without tuning hyperparameters.

**Strengths:**

1. The idea of using multiple frequencies to mitigate PBT’s greediness is innovative and addresses a significant limitation in existing methods.
2.  The asymmetric migration process is a practical solution that could be applied to other population-based methods to improve their performance such as PB2.

**Weaknesses:**

1. All experiments were conducted on several locomotion tasks within the reinforcement learning context, which I believe is unacceptable. HPO has too many application scenarios, and I strongly recommend increasing the variety, such as the experiments in PB2 and PB2-Mix.

2. Other HPO methods (such as Bayesian optimization, e.g.,[1-2] ) are also recommended for comparison. Their experiments are also should be considered to run.

[1] Deep power laws for hyperparameter optimization.

[2] In-Context Freeze-Thaw Bayesian Optimization for Hyperparameter Optimization.

**Questions:**

None.

---

> ### Author Response · Authors · 2024-11-25
> **About our claims**
>
> Thank you for your review of our work, we appreciate that you found our proposed method innovative and motivated.
>
> Regarding the points you noted in your *Weaknesses* section, we would like to re-state the contributions of our work. Then we will proceed to answer on the you two points you outlined.
>
>
> **Our work does not claim to come up with a state-of-the-art HPO method**. Such a statement would indeed require much more benchmarking. The motivation of our work, is that PBT is a highly popular method for RL thanks of its nice properties (dynamic adaptation and variance-exploitation), but its greediness pitfall makes it weak for long-term performance. The claim of our work is that we:
>
> 1. Analyse how this main limitation of PBT can be addressed through the notion of evolution frequency.
> 2. Propose an experimental setup that exhibits clearly PBT's greediness (through the use of the billion steps scale, and the usage of random search baseline)
> 3. Show how MF-PBT is able to overcome the greediness issue.
> 4. Perform ablative studies to better understand the phenomenon.
>
> To this regard, we consider that our experimental setup is valid to support our claims.
>
> Beside this clarification about our claims and our experimental setup, we would like to provide further arguments regarding your remarks.

---

> ### Author Response · Authors · 2024-11-25
> **About the two identified weaknesses**
>
> > All experiments were conducted on several locomotion tasks within the reinforcement learning context, which I believe is unacceptable. HPO has too many application scenarios, and I strongly recommend increasing the variety, such as the experiments in PB2 and PB2-Mix.
>
> We designed MF-PBT without making assumptions on the nature of the learning problem (i.e. task or RL algorithm), because we want the method to be general. We understand that adding a wider variety of tasks in our experiments would indeed make our paper stronger by demonstrating the effective generality of our method.
>
> However we do not agree with the statement about our experimental setting being "unacceptable" for the following reasons:
>
>
> 1. This setting was notably used in SEARL (ICLR 2021), [Eimer et al., 2023](https://proceedings.mlr.press/v202/eimer23a.html) (ICML 2023), BG-PBT (AutoML 2022), which are three papers highly related to our proposition.
> 2. MuJoCo (which Brax is based upon) is a very standard benchmark for RL.
> 3. The five selected environments (Hopper, Ant, HalfCheetah, Walker2D and Humanoid) have already quite different properties. The fact that MF-PBT performs with the same hyperparameters on all those environments is a suggestion of its robustness.
>
>
> The main motivations behind the choice of evaluating our method on Brax are:
>
> - Its huge speed, that enabled us to work on the billion steps scale, which is necessary to make our point on greediness; while making thorough ablative studies on seven random seeds.
> - We are convinced that Brax is to become the next standard benchmark for RL, and it as already been used in [Eimer et al., 2023](https://proceedings.mlr.press/v202/eimer23a.html) and [Wan et al., 2022](https://openreview.net/pdf?id=HW4-ZaHUg5), which are two HPO for RL papers related to our proposition.
>
>
> > Other HPO methods (such as Bayesian optimization, e.g.,[1-2] ) are also recommended for comparison. Their experiments are also should be considered to run.
>
> We would like to answer this point with the same arguments we provided to HjG6 in [this comment](https://openreview.net/forum?id=VLdZkq9xsd&noteId=oFy6PMMMXf):
>
> The Bayesian Optimization (BO) literature pose the HPO problem in the following way: given a fixed computational budget $B$, find the **best performing hyperparameter configuration** possible. The philosophy is that once this fixed configuration is found, practitioners can reuse it to train their model.
>
> Population-based methods (PBT, PB2, SEARL, MF-PBT...) frame the problem as find the **best performing agent possible**, and the budget rather consists of parallel computing abilities (GPU memory in our specific case).
>
> Interestingly, our experiments about variance-exploitation (section 4.3) show that, even when using already tuned hyperparameters, MF-PBT can improve training performance by leveraging RL's intrinsic variance. We believe it further motivates the distinction between the two formulations of HPO.
>
> With respect to this fundamental distinction, we do not believe it makes sense to compare PBT-style and BO-methods. Indeed, regarding how the problem is framed (in terms of budget, hyperparameter search space...), either one or the other class of methods would be advantaged.
>
> Additionally, we believe our experimental setting to be quite unfair to BO-methods. As mentioned in section 4, we tune only two hyperparameters (learning rate and entropy coefficient), and all the remaining hyperparameters (gamma, batch sizes...) are taken from the already-tuned configurations proposed by Brax. This setting makes Random Search quite representative of the results achievable by methods yielding fixed hyperparameters configurations. The fact that MF-PBT performs better than the 7 seeds of 32-agents RS (Figure. 1), indicates it would probably also outperform BO methods (in this setting).
>
> We hope these clarifications offer a fuller understanding of our methodology and its objective, and remain open to further discuss on additional points.

---

> > ### Comment · Reviewer_Xs8X · 2024-11-26
> >
> > Thank you for your detailed reply.
> >
> > I acknowledge the motivation and contribution of MF-PBT, particularly as it serves as a general framework for PBT without making assumptions about the nature of the learning problem. That is why I hope to see its general capabilities in **this** work, rather than leaving this as future work, especially considering that many PBT studies have been conducted in contexts beyond RL. At least adding one or two experiments would help demonstrate the "general" nature to some extent.
> >
> > My main concern still exists (my initial score is closer to 4, but this year's options do not allow for that), so I will hold my current score to reflect this. I believe this paper has significant room for improvement (which is even not hard to achieve in my opinion) to enhance it and the current version is not well prepared for publication.

---

> > > ### Author Response · Authors · 2024-12-02
> > > **On our experimental design**
> > >
> > > Thank you for considering our response and acknowledging the contribution of MF-PBT. While we understand your position on our experimental setup, we would like to take this opportunity to clarify the rationale behind our design choices.
> > >
> > >
> > > **Focusing on Reinforcement Learning**:
> > >
> > > While the original PBT paper (Jaderberg et al. 2017) introduced PBT as a general HPO method, its greatest impact has been within the RL community. This is largely due to two factors:
> > >
> > > 1. As noted in Sections 1 and 2 of our paper, PBT's strengths—dynamic adaptation and variance-exploitation—are particularly suited to RL's defining characteristics, such as non-stationarity and intrinsic variance.
> > > 2.  Additionally, RL for control tasks typically uses small neural networks (e.g., 4-layer MLPs), making it computationally feasible to train 32 networks in parallel.  This is not the case with the standard architectures in CV and NLP.
> > >
> > > > especially considering that many PBT studies have been conducted in contexts beyond RL
> > >
> > > This claim is not accurate. With the exception of FIRE PBT (which we address below), all extensions to PBT (SEARL, PB2, PB2-Mix, BG-PBT) cited in Section 2 focus exclusively on RL, primarily for the reasons outlined above.
> > >
> > > FIRE PBT's (Dalibard & Jaderberg 2021) experiment on ImageNet highlights why population-based methods are challenging to scale outside RL. Each run required a population of 50 ResNet50 models trained in parallel, amounting to 200 TPU v3-core-days per seed. Scaling this to seven random seeds across three baselines would cost approximately 4,200 TPU-days (~ $120,000), illustrating the prohibitive resource requirements for such experiments.
> > >
> > >
> > > PBT (Jaderberg et al. 2017) introduced a new framework and emphasized its generality by including experiments outside RL. However, as subsequent works have primarily focused on RL, and since we frame MF-PBT within this family, we chose not to experiment beyond RL.
> > >
> > >
> > > **MuJoCo is a sufficient benchmark for RL**:
> > >
> > > As we explained in our previous answer, MuJoCo is a well-established benchmark for RL algorithms. Some of the most important RL papers, such as [SAC](https://proceedings.mlr.press/v80/haarnoja18b/haarnoja18b.pdf), [PPO](https://arxiv.org/pdf/1707.06347) or [TD3](https://proceedings.mlr.press/v80/fujimoto18a/fujimoto18a.pdf) did perform experiments only on MuJoCo tasks.
> > >
> > > One of the main motivation is that control tasks are computationally cheaper (small neural networks), while remaining difficult to solve (as opposed to naive tasks such as CartPole). This allow to run experiments on multiple seeds to robustify the results.
> > >
> > > This consideration is particularly relevant for MF-PBT: to evidence the greediness of PBT and demonstrate that MF-PBT is able to overcome it, it requires to perform extended trainings on tasks that possess multiple optima, making the Brax framework perfectly suitable for our experiments.
> > >
> > > This exact same framework was used in SEARL and BG-PBT. PB2 added an experiment on two envs from Atari, but with a population size of 4 agents, and focusing only on early training. PB2-Mix did only use the ProcGen benchmark, as they wanted to experiment on data augmentation.
> > >
> > >
> > > **PDF updated with an experiment on Pusher**:
> > >
> > > During the redaction of the paper, we also conducted experiments on the *Pusher* environment, which is a sub-task of Brax, where the agent must learn to control a robotic arm. However, the simulation in *Pusher* is way slower than the other environments,and to obtain results for seven random seeds, we capped the experiment at 300 million steps.
> > >
> > > We decided to not include this experiment in the paper primarily due to lack of space, and also due to smaller training timescale that wasn't consistent with the other environments.
> > >
> > > However your first comment on the fact that our experiments only include locomotion tasks led us to reconsider the value of this experiment. We thus decided to include it in our appendix, in figure 8 of the updated pdf.
> > >
> > >
> > > We hope these clarifications provide a stronger justification for the experimental design of our work and address your concerns about its readiness for publication.

---

### Note · Authors · 2025-05-31

I have read and agree with the venue's withdrawal policy on behalf of myself and my co-authors.

---

### Meta-Review · Area_Chair_7Thu · 2024-12-19

**Metareview:**

This is a nice idea and has the ingredients of a good paper. The main concerns are the lack of comparisons to other approaches (e.g. PB2, or MF methods), and the lack of diversity of tasks. I understand the authors use historical precedent to focus on RL, but then there needs to be more emphasis on what this new approach really enables. At present, it may be simply solving problems addressed in BGPBT or PB2 and so should be compared to those, or attempted to combine with those, to provide a clearer picture and become a truly valuable contribution that meets the bar for ICLR.

**Additional Comments On Reviewer Discussion:**

There was a discussion on the claims of the authors, as reviewers were concerned this was a general HPO algorithm. It is clearly focused on AutoRL and that makes sense, and seeking to address a single problem for PBT rather than be SOTA. That being said, there are other PBT variants that have tried to address the same issues, so there should be some comparison here.

---

### Decision · Program_Chairs · 2025-01-22

Reject